# Real-Time Integration of Segmentation Techniques for Reduction of False Positive Rates in Fire Plume Detection Systems during Forest Fires

**Leonardo Martins** [1,2,†] , **Federico Guede-Fernández** [1,2,3,†] , **Rui Valente de Almeida** [1,2,†] , **Hugo Gamboa** [1,3] and **Pedro Vieira** [1,2,*]

1    Physics Department, NOVA School of Science and Technology, Campus de Caparica, 2829-516 Caparica, Portugal; l.martins@campus.fct.unl.pt (L.M.); f.fernandez@fct.unl.pt (F.G.-F.); rui.almeida@future-compta.com (R.V.d.A.); hgamboa@fct.unl.pt (H.G.)
2    Future Compta S.A, 11495-190 Alges, Portugal
3    LIBPhys (Laboratory for Instrumentation, Biomedical Engineering and Radiation Physics), NOVA School of Science and Technology, Campus de Caparica, 2829-516 Caparica, Portugal
*    Correspondence: pmv@fct.unl.pt
†    These authors contributed equally to this work.

**Abstract:** Governmental offices are still highly concerned with controlling the escalation of forest fires due to their social, environmental and economic consequences. This paper presents new developments to a previously implemented system for the classification of smoke columns with object detection and a deep learning-based approach. The study focuses on identifying and correcting several False Positive cases while only obtaining a small reduction of the True Positives. Our approach was based on using an instance segmentation algorithm to obtain the shape, color and spectral features of the object. An ensemble of Machine Learning (ML) algorithms was then used to further identify smoke objects, obtaining a removal of around 95% of the False Positives, with a reduction to 88.7% (from 93.0%) of the detection rate on 29 newly acquired daily sequences. This model was also compared with 32 smoke sequences of the public HPWREN dataset and a dataset of 75 sequences attaining 9.6 and 6.5 min, respectively, for the average time elapsed from the fire ignition and the first smoke detection.

**Keywords:** fire detection; smoke plume; deep learning; image segmentation

## 1. Introduction

According to reports regarding forest fires in Europe, the Middle East, and North Africa, 2019 was the worst in terms of total burnt area, and despite the increase of regulation programs and financing levels in the EU countries, about 340,000 hectares (ha) were still burnt in the EU in 2020 [1], making wildfires still a large problem, causing natural damage, such as deforestation and defaunation and even human deaths with an estimated mean burned area of over 4 million km² around the globe [2]. Prescribed burning is still being used as one of the methods to suppress the advancement of wildfires [3], which is an activity that is usually regulated by governmental policies [4].

Wildfires have been associated to increases of ambient levels of particulate matter [5], including tropospheric ozone levels [6], with direct and indirect links to ecological and health damages [7]. Wildfires have also been found to only have negligible benefits to some plant and animal species while leaving irreparable damage to other species [8], due to their sudden appearance and fast spreading, which in turns favors the implementation of real-time monitoring systems to prevent their escalation.

Recent reviews have shown the long-term importance of investing (while having high short-term costs) in detection innovations [4], with those ranging from signal-based to image-based [9,10]. While applications have widely different mechanisms of functioning,

most of these applications have implemented some sort of Machine Learning (ML) methods or upgraded into the use of Deep Learning (DL) [10,11]. Real-time fire monitoring systems can be separated by their physical implementation: terrestrial-based, aerial-based and satellite-based, each with their own pros and cons [10].

The employment of satellites such as MODIS (Moderate Resolution Imaging Spectroradiometer) has been extensively used for earth monitoring activities [12]. Satellites can be used for the classification of smoke [13] (SmokeNet platform) and fire (using the Inception-v3 architecture) [14]. In terms of aerial techniques, the largest use is the application of Unmanned Aerial Vehicles (UAVs), which have been extensively reviewed [15] and can also benefit from Artificial Intelligence (AI) techniques for fire detection [16]

Finally, regarding terrestrial services, the main focus of our research [17–19], they have mostly evolved into the use of surveillance fixed lookouts, such as the one implemented in [20], which uses DL and feature extraction methods. Some reported works use proprietary technology and have been already converted in real-time commercial systems such as FireWatch and ForestWatch [21] and FireScout by AlcheraX Inc., which also incorporate cloud-based machine vision AI to detect wildfires [22].

The deployment of image-based applications in real-time context poses several challenges such as different backgrounds, obstacles in the angle of vision, artifacts such sun flares, fog, clouds or animals while also having to deal with the variability of the shape, motion and color of the smoke and flames [23].

The above problems were also identified in our previous work, where a model based on object detection and DL for early detection of smoke plumes was presented with high accuracy [19]. The aim of this work is to lessen these drawbacks by proposing a hybrid model that combines several strategies, such as DL algorithms with instance segmentation, feature extraction and traditional Machine Learning classification techniques. The main impact of this study is to address the problem that every alarm still needs to be confirmed by a manual operator from the forest protection services, and persistent false positives could lead to a human tendency to start ignoring some of those alarms.

## 2. Related Work

As mentioned above, our research work focuses on the use of optical sensors implemented in a terrestrial fixed lookout system, using both DL and traditional ML methods to classify objects containing smoke and not containing smoke plumes [19]. This section presents research works with similar technologies applied to the identification of indoor and outdoor fires. This section also identifies studies that have dealt with correcting a real-time persisting false positive rate.

It is important to remark that sensor technologies applied to the identification of fire and smoke have received several extensive reviews [9–11,23]. A recent review on vision-based outdoor smoke detection was able to identify 126 papers relevant to this problematic, highlighting the use of ML and DL algorithms for the reduction of the false alarm rate (FAR) during the detection of outdoor smoke plumes [24].

A study has developed a Bayesian network-based information fusion system connected with deep neural networks to robustly detect fire objects, combining environmental information (e.g., humidity) with visual information (e.g., location recognition) [25]. The study then used Faster Region-based Convolutional Neural Network (R-CNN) to detect the suspected regions of fire, and long short-term memory (LSTM) accumulates the local features within the bounding boxes to classify the object [25]. Their main reduction of false positive detections was based on a hard negative example mining including images of clouds, chimney smoke, lighting lamp, and steam from several data sources, including the Flickr-Fire dataset compiled in [26]. Their majority voting of short-term decisions is taken to make the decision reliable in a long-term period [25].

Another work has reduced the number of False Positives by using spatial and temporal features of smoke and the fire flame, such as frame difference, color, similarity, wavelet transform, coefficient of variation, and MSE [27]. Their initial step was based on detecting

the object movement, by calculating the global frame similarity and the mean square error [27]. The regions of interest were extracted using a R-CNN algorithm, presenting a reduction of 99.9% of the number of false positives [27]. While their dataset was based on mainly factory and office indoor settings [27], and not on outdoor wildfires, their strategies might be able to be transposed to that problematic.

Another study also used CNNs combined with conventional image processing to segment the regions with flames by extracting their color features [28]. Afterwards, feature maps such as shape and texture were extracted with a convolutional neural network based on an adaptive pooling method [28]. Accuracy increased from around 80% to over 90% when using the segmented fire area image as the training sample instead of the original image [28]. While the detection of fire flames is mostly used in small-range sensors, they can also be implemented in the detection of late-stage fires, as the detection of the smoke plumes can be a better indicator of early-stage fires.

A distinct approach by [29] combined dynamic and static feature extraction, using deep learning (based on a Caffemodel) and the degree of irregularity combined with an adaptive threshold. They reduced false positives by dividing the image into smaller grids and recording temporal information of the fire and smoke [29]. An algorithm based on three stages, namely a spatial features extraction network, a Bidirectional LSTM and a temporal attention subnetwork were used to detect wildfires at an initial ignition, reducing the FAR over previous image-based DL models [30].

Another CNN-based algorithm was shown to achieve better results than the YOLOv3 while detecting small indoor and outdoor flames [31]. They used appearance-based pre-processing techniques such as HSV color conversion and Harris Corner Detection to reduce the FAR, instead of using directly the acquired images as input. This approach can also be extended to detect smoke columns [31].

Finally, it is also important to make comparisons regarding satellite and aerial-based systems with similar case studies. A research work used transfer learning based on the Inception-v3 platform, and it was able to classify datasets into fire and non-fire images, mainly by detecting the presence of smoke. They were also able to reduce the detection of false positives by using local binary patterns [14]. In terms of aerial systems, some studies using UAVs were able to detect forest fires also using DL approaches based on CNNs with the help of the YOLOv3 platform. While their accuracy was around 83%, they showed a vast potential due to their high temporal resolution [32].

## 3. Materials and Methods

### 3.1. Workflow

As mentioned, this research is a direct continuation of a previous work [19]; therefore, this work uses the Detectron2 platform as the foundation for the workflow of this research work, as shown in Figure 1. The Detectron2 platform was used for the objection detection phase (i.e., the first smoke classification algorithm) and for the object segmentation phase. Feature extraction methods and Machine Learning algorithms were also added to the second smoke classification algorithm (see Figure 1).

As can be seen, some of the dataset and methods are intertwined with the previous work [19], while significant new data and methods were added. The Detectron2 platform, which is implemented based on the open source machine learning framework Pytorch, has been continuously updated by the Facebook AI Research [33]. While this platform was initially implemented for face detection tasks, it has now evolved and been in several other projects, including outdoor semantic examples, mainly focused on objects such as traffic signs, bicycles, cars, etc. [34]. Detectron2 can be also be used to train various state-of-the-art base models (e.g., Fast R-CNN, Faster R-CNN and RetinaNet) for COCO Object Detection and COCO Instance Segmentation tasks (e.g., Mask R-CNN) from their Model Zoo [33], and it has several backbone combinations, namely Feature Pyramid Network (FPN), C4 and DC5, which have been found to achieve a very high accuracy in object detection tasks [35].

The following sub-sections will provide a succinct summary of the previous methods and dataset that still apply in this work and a detailed description of the newly added datasets and methods.

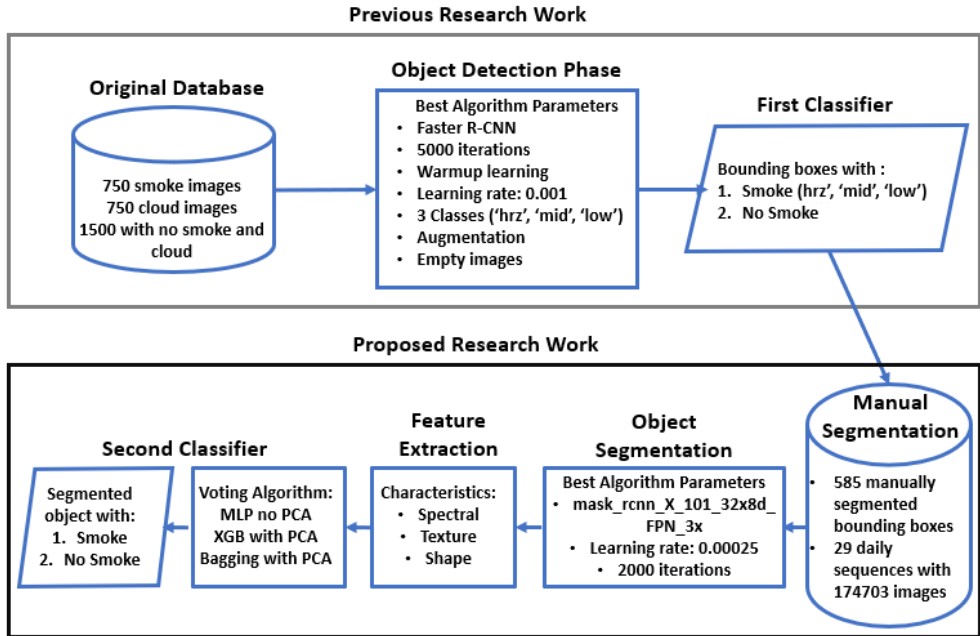

**Figure 1.** Workflow and overview of the best parameters of each step of the research work.

### 3.2. Dataset

The dataset from the previous work [19] was used to make the final comparison with the newly developed algorithms (see Original Database in Figure 1), which was performed in ten different systems installed all around the Peneda-Gerês National Park in Portugal, which were recently decommissioned. Two new locations (near the cities of Valongo and Proença) were thus installed in the year of 2021 and were used to create real-time daily sequences during the summer and fall season of 2021. The acquisition system is based on an optical camera which is mounted on a fixed post and has a pan and tilt movement controlled remotely by a server; namely, a bi-spectrum temperature measurement camera IQinVision IQeye 7 Series (IQ762WI-V6) was used for the acquisitions. Camera specifications for each acquisition were the following: an image sensor of 1/3″ CMOS sensor, a 12–40 mm telephoto lens with a 18° wide and 9° tele-oscillatory ventilation and with an effective resolution of 1080 p. While these new datasets were acquired with the same acquisition system as in our previous work [18,19], they have different ecosystems, as the Valongo system is in the north of Portugal with a large and dense vegetation area but also some urban areas, while Proença is in the south of Portugal with sparse vegetation area and almost no urban areas.

Each daily sequence was only analyzed during the day, as the implemented algorithms only work in visible light. The azimuth of the camera rotated with a pre-defined angle at a fixed time rate in order to span the 360° around it is fixed post, completing a complete scan in around 5 min (rotation is not done at a full 360°, as it rotates back and forth until it reaches a stopping point).

During the previous research work [19], the original dataset (see Figure 1) was divided into 750 images with smoke, 750 images containing clouds and 1500 images not containing significant objects (clouds and smoke objects). Furthermore, the images with smoke were further labeled with three sub-classes of objects namely: *hrz*, *mid* and *low*, with the first ones defined as smoke that is visible above the horizon level with a predominant cloud background and can for example be mistaken easily with clouds, the seconds defined as smoke present in the middle of the image, with a predominant land background and the

third defined as smoke that occurs near the acquisition system and can occupy a large proportion of the image.

Examples of detected fires are shown in the Results section. This labeling process was performed with the OpenLabeler platform [36], initially saving into a PASCAL VOC XML format and converted into a JSON file in COCO format to be read by the Detectron2 platform.

The manual segmentation process (see workflow in Figure 1) was performed with the Image Processing Toolbox from the MATLAB® platform. Segmentation was done with the polygon creation feature in order to be processed by the Detectron2 platform. The annotation of the visible objects was done inside the previously detected bounding boxes and tagged with a smoke name and into a JSON file. Figure 2 shows three examples of manually segmented fires and their final polygons.

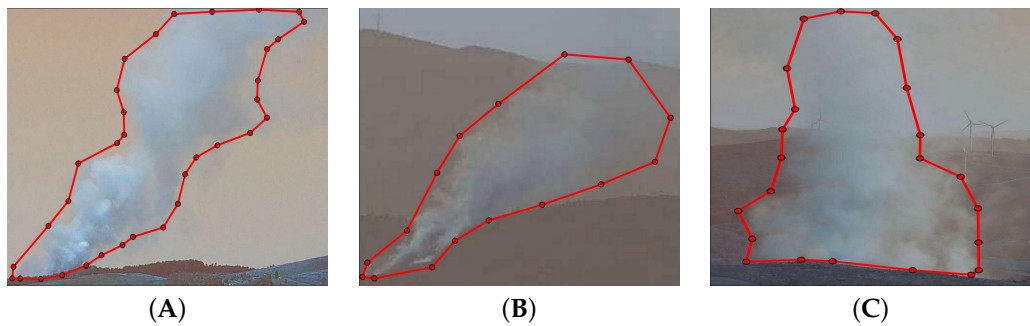

**Figure 2.** Different types of segmented fire smoke columns. (**A**) Example of smoke labeled as *hrz*. (**B**) Smoke labeled as *mid*. (**C**) Smoke labeled as *low*.

The training of the object segmentation algorithm was done with 29 new daily sequences (see Figure 1), with a total number of 174,703 images, for a total of 10,333 detections using the previously implemented algorithm. A breakdown of the number of images and total detections for each daily sequences is presented in the Supplementary Table S1.

*3.3. Algorithms*

3.3.1. Smoke Segmentation with Deep Learning Algorithms

In this paper, the Detectron2 framework has been used to build the smoke detection model based on a two-stage object detection architecture and then afterwards implement an instance segmentation algorithm to segment the obtained bounding box, as seen in Figure 1. The first detector is fully described in [19]. In summary, the initial training was done with two base models (RetinaNet and Faster R-CNN) with the original ResNet-50 backbone combined with the FPN and using the initialized weights of their pre-trained architecture on the MS COCO dataset ($3\times$, $\sim$37 COCO epochs) [33]. Two main parameters (number of iterations and learning rate strategy) were evaluated during the training phase, ending up with the best parameter of warm-up constant strategy (WUP) with a learning rate of 0.001 and with 5000 iterations (see Figure 1).

For the instance segmentation training, we carried out 27 training sessions by modifying three main combinations of parameters, namely the learning value (0.001, 0.0001, 0.00025), the maximum iterations value (1000, 2000, 3000) and the Base Model ((R_50_FPN_3x, R_101_FPN_3x, X_101_32x8d_FPN_3x), as defined in the Detectron2 Model Zoo [33]. The choosing of the final parameters was based on the Average Precision metrics (as defined in the next sub-section). It is remarked that the segmentation is only performed inside the bounding boxes provided by the first classification algorithm.

3.3.2. Feature Extraction

In this research work, after the segmentation process was completed, we extracted spectral-based features (average and standard deviation of the Blue, Green and Red channel), texture-based features, namely (matrix of co-occurrences of grays (GLCM), contrast, dissimilarity, homogeneity, energy, correlation and second angular moment) and finally

shape-based features (eccentricity, area, compactness, density, roundness, rectangular fit, elliptical fit and asymmetry). This feature extraction process was done using a similar methodology as the one applied in [37] for objects detected in images acquired by UAVs.

The spectral-based features were calculated only inside the segmented object, instead of using zeros outside of the segmented object; we use the 'not a number' feature of the mean and standard deviation calculation. The GLCM Texture Features were directly calculated using the scikit-image software, which implements and reproduces the Haralick analysis [38]. The following shape features were directly extracted with the *regionprops* function (which measure the properties of labeled image regions) [39]: eccentricity and solidity (density). The other features were calculated [40]: using parameters extracted directly from the *regionprops* function, namely for compactness (area and perimeter), roundness (area and major axis length), rectangular and elliptical Fits (area, bounding box area, major and minor axis length), as can be observed in Equations (1)–(4):

$$\text{Compactness} = \frac{4 * \pi * \text{Area}}{\text{Perimeter}^2} \tag{1}$$

$$\text{Roundness} = \frac{4 * \text{Area}}{\pi * \text{Major Axis Length}^2} \tag{2}$$

$$\text{Rectangular Fit} = \frac{\text{Area}}{\text{Bounding Box Area}} \tag{3}$$

$$\text{Elliptical Fit} = \frac{\text{Area}}{\pi * \text{Major Axis Length} * \text{Minor Axis Length}} \tag{4}$$

### 3.3.3. Smoke Classification with Machine Learning Algorithms

Finally, the implementation of the second classifier of smoke and non-smoke images was based on the extracted feature inputs. The implementation of all the Machine Learning algorithms was done on the scikit-learn Python package [41], which provides a wide range of state-of-the-art machine learning algorithms, and it allows the integration of supervised models and the use of principal component analysis (PCA) to reduce the number of input variables using Singular Value Decomposition. In order to perform PCA reduction of the initial number of features, 21, the feature union estimator was used to perform a grid search to check all the integer possibilities from 1 to 15 [41].

The Machine Learning algorithms that were tested have all been extensively reviewed in Remote Sensing applications, such as "Decision Trees (DT)", "Random Forests (RF)", "Boosted" trees such as "AdaBoost", "XGBoost", and also the use of "Bagging" methods in Decision Trees, "Multi-Layer Perceptrons (MLP)" and "k-Nearest-Neighbor (kNN)" [42]. We also used two other popular Machine Learning algorithms for remote sensing applications, namely "Linear Discriminant Analysis (LDA)" [43] and "Naïve Bayes Classifier (NB)". For each Machine Learning algorithm, a grid search was performed to find the hyper-parameters for our supervised problem, as shown in Table 1. Finally, a voting ensemble algorithm with soft labels [44] (predicting the class based on the sums of each predicted probability), based on the three best performing algorithms and their chosen parameters (see Section 4.2) was also trained and evaluated.

Testing and training of the object detection and segmentation models were all performed on a computer configured as follows: the CPU was an Intel i7 9700k at 3.6 GHz with 8 cores, the graphics card was a dual NVIDIA RTX2080, the RAM memory was 64 GB, and there was a hard disk of 2 TB. The software environment was as follows: the operating system was Ubuntu 18.0.5, the programming language was Python 3.8.5 and the main Python software libraries were: pytorch (v.1.7.1), torchvision (v. 0.8.2), opencv-python (v. 4.4.0.46), Detectron2 (v. 0.3), albumentations (v. 0.5.2) and numpy (v. 1.19.2), wandb (v. 0.10.12), pandas (v. 1.3.4), scikit-learn (v. 1.0.2), scikit-image (v. 0.20.0), imblearn (v. 0.8.1), xgboost (v. 1.5.0), and seaborn (v. 0.11.2).

**Table 1.** Searched hyper-parameters during the training of the smoke classification algorithm.

| Algorithms | Searched Parameters | Values or Variables |
|---|---|---|
| KNN | Number of nNighbors<br>Power Parameter | ['1', '2', '3', '4', '5', '6']<br>['1', '2'] |
| DT | Criterion | ['gini', 'entropy'] |
| LDA | Solver | ['svd', 'lsqr', 'eigen'] |
| NB | Smoothing Variable | ['0.00000001', '0.000000001', '0.00000001'] |
| RF | Number of Estimators<br>Maximum Depth | ['10', '50', '100', '500']<br>['4', '6', '8', '10', '12', '14'] |
| AdaBoost | Number of Estimators<br>Learning Rate | ['10', '50', '100']<br>['0.0001', '0.001', '0.01', '0.1', '1.0'] |
| XGBoost | Default Parameters Used | ['Not applicable'] |
| Bagging | Number of Estimators | ['10', '50', '100', '500'] |
| MLP | Size of the Hidden Layer<br>Max Number of Iterations<br>Activation Function<br>Solver Method<br>Alpha<br>Learning Rate<br>Learning Rate Initial Value | ['(150, 100, 50)', '(120, 80, 40)', '(100, 50, 30)']<br>['100', '500', '1000']<br>['tanh', 'relu']<br>['sgd', 'adam']<br>['0.0001', '0.05']<br>['constant', 'adaptive']<br>['0.001', '0.0005'] |

*3.4. Performance Evaluation*

The performance evaluation of the training and testing of the instance segmentation algorithm is done with the Average Precision (AP) and Average Recall (AR). When evaluating the detection of objects and defined segments, it is also necessary to define the intersection over union (IoU) metric, which calculates the ratio of the intersection area of the bounding boxes over the area of the union of the bounding boxes, determining the overlap between the obtained bounding boxes and the labeled ground-truth. Using the IoU, it is possible to confirm if the detected object is true or false, by setting the IoU into a fixed threshold value. In this specific analysis, we used the AP and AR metric (see Equation (5)), corresponding to the average AP for IoU from 0.5 to 0.95 with step size of 0.05 for both the bounding box (bb) calculation and the segmentation (sg). An AP50 corresponding to an IoU of 0.50 and AP75 corresponding to an IoU of 0.75 are also presented.

$$A\{P,R\} = \frac{\sum_{i \in \{0.5,0.55,...0.95\}} A\{P,R\}_i}{10} \tag{5}$$

The evaluation of the proposed methodology is based on the manual confirmation of the results of the 29 daily sequences, which are based on the results of the previous classification and the proposed classification algorithm (see Figure 1). To calculate the Sensitivity and Specificity metrics, we need to identify the True Positives (TP) and False Negatives (FN), which are the number of images manually annotated as smoke and which the classifier identified as smoke and non-smoke, respectively. The objects identified as True Negatives (TN) and False Positives (FP) are the boxes manually annotated as no-smoke and then classified as no-smoke and smoke, respectively. The Sensitivity, Specificity, Accuracy and F1-Score metrics (obtained from the aforementioned TP, FP and FN) were macro-average, meaning that they were calculated by taking the arithmetic mean (aka unweighted mean) of all the per-class scores. The classification performance scores in validation and testing are expressed in Equations (6)–(9):

$$\text{Sensitivity} = \frac{\text{TP}}{\text{TP} + \text{FN}} \tag{6}$$

$$\text{Specificity} = \frac{\text{TN}}{\text{TN} + \text{FP}} \tag{7}$$

$$\text{Accuracy} = \frac{\text{TP} + \text{TN}}{\text{TP} + \text{TN} + \text{FP} + \text{FN}} \tag{8}$$

$$\text{F1-Score} = \frac{\text{TP}}{\text{TP} + \frac{1}{2}(\text{FP} + \text{FN})} \tag{9}$$

We also calculated the time elapsed to detect the first fire in a daily sequence using the previous model [19], the method previously compared with [45] and the model proposed here. The detection rate (DTR) metric was defined to assess the sequences where at least one image was detected, showing the smoke detection performance over time. This metric is calculated for each sequence by checking if at least one true smoke detection was made during the long presence of a smoke. In this case, we computed it as 1 if at least one image of the smoke was identified or 0 if no image during the entire time-series was identified.

## 4. Results and Discussion

As mentioned in Section 3.3, the previously developed classification [19] algorithm was deployed in 2021 in two new locations (Valongo and Proença) for daily performance evaluation in real time. On average, 6000 images are obtained every day, having around a 5% persistent False Positive rate, which account for around 300 daily reports. Examples of these False Positives were identified to be, for example, poles in the front of the camera, water drops or sun reflections, which eventually produce false smoke detections on a consistent basis. Figure 3 shows four examples of this type of False Positive. Since these reports were being directly sent to be analyzed by a manual operator from the forest protection services, it was decided that the number of False Positives needed to be reduced further without severely compromising the detection of True Positives.

The solution proposed here uses a hybrid model, which combined the previous deep learning algorithm with instance segmentation on the bounding boxes produced by that algorithm. The segmentation results are then used to extract several features (shape, spectral and texture-based) that can finally be inputted for the final smoke classification algorithm based on traditional Machine Learning methods.

To evaluate the performance of the developed algorithms, 29 daily sequences were chosen randomly for a total of 174,703 images. The total number of acquired images for each day can be checked on Table S1 in the Supplementary Material, and the daily amount differs because only daylight images are analyzed in this process. Out of these 29 sequences, only 6 had confirmed fires, which was later verified by the Portuguese Forest Protection Services. As can be checked, the initial algorithm had 10,333 detections, from which 10,188 were manually verified to be false positives, as shown in Figure 3.

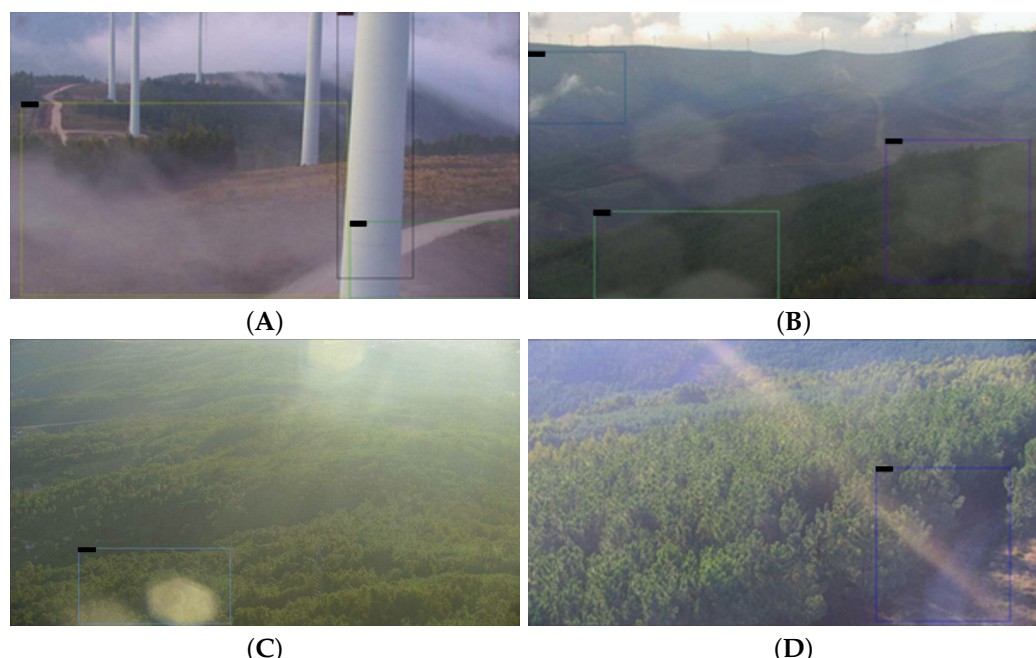

**Figure 3.** Examples of false positives detected with over 50% probabilities. In example (**A**), bounding boxes are shown with detections of a pole and fog in front of the camera. Example (**B**) shows three bounding boxes, with only one being truly positive, while the other two are water drops in the lens. In example (**C**), there is a heavy fog in front of the camera that causes a False Positive to be produced. Example (**D**) shows a reflection of the sun which, when superimposed on a road, resulted in a False Positive.

### 4.1. Smoke Segmentation

The initial step to implement the instance segmentation algorithm was to manually segment 585 images of smoke plumes, as seen in Figure 2 inside the bounding box. As mentioned in the Methods and Algorithms section, it was decided to use 29 daily training sessions, which were carried out each with different combinations of parameters; namely, we changed the learning value (0.001, 0.0001, 0.00025), the maximum iterations value (1000, 2000, 3000), and the Base Model ((R_50_FPN_3x, R_101_FPN_3x, X_101_32x8d_FPN_3x), as defined in the Detectron2 Model Zoo [33], with each case identified in Table S2 in the Supplementary Material. The two best case scenarios (case 20 and 06) and the worst case results (Case 07 and 16) can be observed in Table 2.

**Table 2.** Values (in percentage) for the Bounding Box (bb) and Segmentation (seg) Average Precision (normal, 50 and 5) and Average Recall (AR) for the Segmentation, for the best case scenarios (20 and 06) and worst case scenarios (07 and 16).

|         | Case 20 | Case 06 | Case 07 | Case 16 |
|---------|---------|---------|---------|---------|
| bbAP    | 48.88   | 48.06   | 21.48   | 25.41   |
| bbAP50  | 89.75   | 93.22   | 62.78   | 70.57   |
| bbAP75  | 34.67   | 26.62   | 2.44    | 5.77    |
| segAP   | 38.63   | 37.61   | 17.14   | 20.12   |
| segAP50 | 83.22   | 84.36   | 57.74   | 67.57   |
| segAP75 | 34.67   | 26.62   | 2.44    | 5.77    |
| segAR   | 47.60   | 45.50   | 33.40   | 36.40   |

The best results were obtained with the following parameters: Best Base Model (mask_rcnn_X_101_32x8d_FPN_3x), learning value with warm-up (0.00025) and 2000 iterations (case 20) and (mask_rcnn_X_101_32x8d_FPN_3x), learning value with warm-up (0.00025) and 2000 iterations (case 06). Table 2 shows the Average Precision (AP), including

the AP50 and AP75 (as defined in the Methods Section) for both the bounding box creation (bb) and the segmentation (seg). The training results are comparable to the literature [46] in the detection of other objects using RetinaNet and FRCNN. Examples of the segmentation of true positives (visible smoke) can be seen in Figure 4.

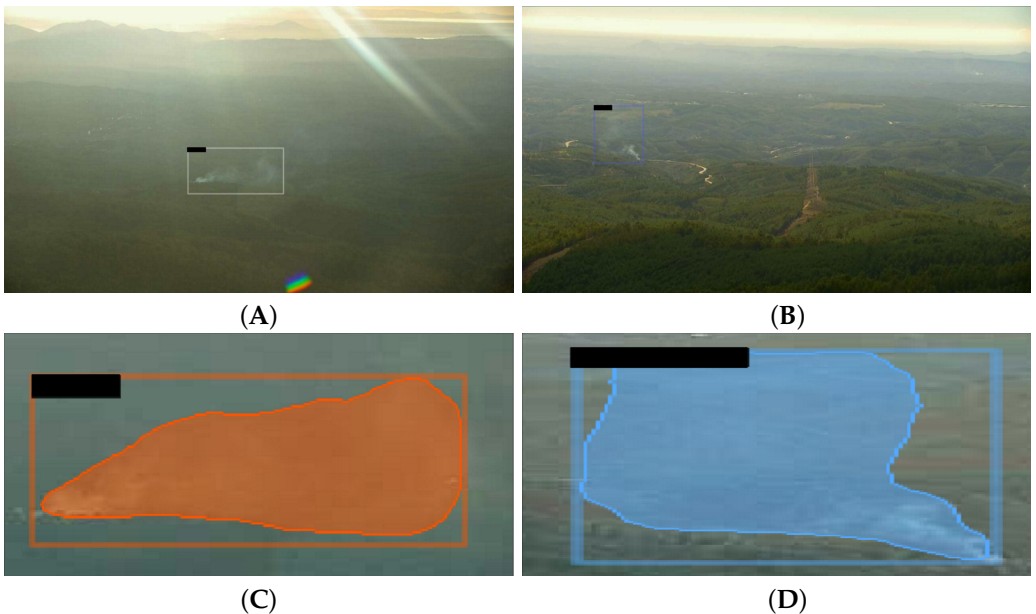

**Figure 4.** Examples of true positives detected and their segmentation outlines. (**A**) shows a detected bounding box (with 89.7%) of a *mid* smoke column and the respective segmentation in (**C**), which was later identified as a smoke object with a probability of 77.5%. (**B**) shows another detected bounding box (with 66.1%) of a *hrz* smoke column and respective segmentation in (**D**), which was later identified as a smoke object with 87.4%.

### 4.2. Smoke Classifier Based on Machine Learning

Each of these input values was trained and tested with traditional Machine Learning algorithms, and the results of each of these algorithms (in terms of their Accuracy) are shown in Table 3.

**Table 3.** F1-Score (Macro) results (in percentage) of the training, for each ML algorithm without PCA and the use of PCA for variable reduction.

| Algorithms | No PCA | With PCA |
|:---:|:---:|:---:|
| MLP | 84.3 | 82.9 |
| Random Forests | 81.3 | 82.9 |
| Bagging | 81.45 | 82.9 |
| LDA | 73.9 | 69.8 |
| Naïve Bayes | 60.13 | 61.8 |
| Decision Trees | 74.2 | 78.0 |
| k-Nearest Neighbor | 57.2 | 56.4 |
| AdaBoost | 76.3 | 78.1 |
| XGBoost | 81.4 | 84.5 |
| Voting Algorithm | 84.6 | |

A voting algorithm was also chosen (see Section 3.3), aggregating three best-performing algorithms in terms of the macro-averaged F1-Score, namely the MLP classifier (without PCA) with the following parameters: 'activation' = tanh, alpha = 0.0001, 'hidden layer sizes' = (150, 100, 50), 'learning rate' = constant, 'learning rate init' = 0.001, 'max iter' = 1000, and 'solver' = adam. The second algorithm was the XGBoost classifier using PCA with

12 features and the Bagging classifier, with 100 estimators and using PCA with nine characteristics. It is noteworthy to mention that the voting algorithm obtained a final accuracy of 84.60%, which is superior to the three algorithms implemented individually. An evaluation of the voting algorithm was carried out, using a 10-fold cross-validation, obtaining an average Precision of $88.4 \pm 1.3\%$, an Accuracy of $88.3 \pm 1.2\%$, a Recall of $95.5 \pm 1.0\%$ and finally an F1-Score of $91.8 \pm 0.9\%$.

Two interesting examples are shown in Figure 5; the first one, in Figure 5A, shows two detections, one incorrect and later rejected (C) and one correct and later accepted (E), maintaining the TP and changed an FP to a TN. The second example, in Figure 5B, shows another two detections, one incorrect (D) and one correct (F), but while this time the correct was maintained as a TP, the incorrect was also maintained as an FP.

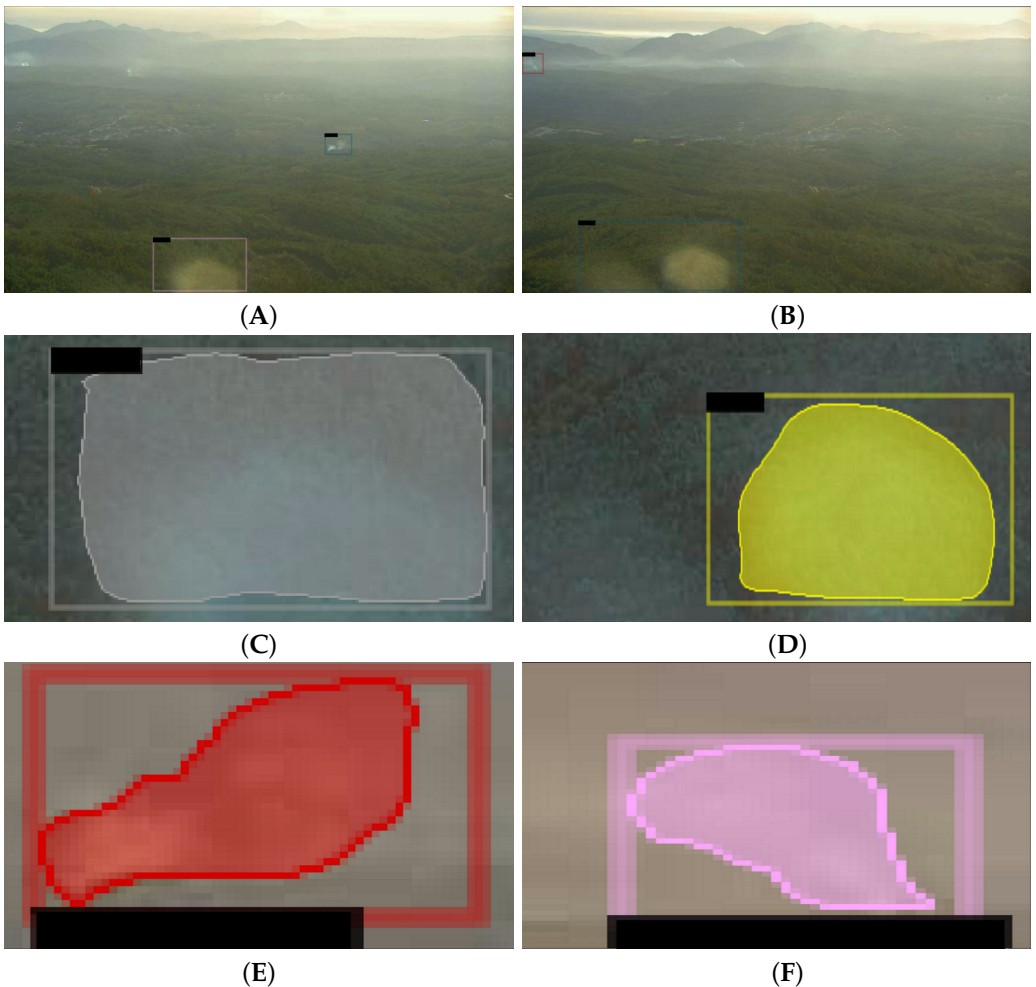

**Figure 5.** Examples of true and false positives detected and their segmentation outlines. Figure (**A**) shows two detected (with the object detection algorithm) bounding box of a *hrz* smoke column and the respective segmentation in (**C**,**E**). The object (detected with 69.1%) in (**C**) was later correctly rejected (with the machine learning algorithm) as a smoke object with a probability of 21.3%, while object (detected with 90.9%) in (**E**) was correctly identified as smoke with a probability of 96.2%. Figure (**B**) shows another double detection of *hrz* objects, and the respective segmentation in (**D**,**F**). The object (detected with 85.0%) in (**D**) was later incorrectly detected as a smoke object with a probability of 84.4%, while the object (detected with 96.9%) in (**E**) was correctly identified as smoke with a probability of 95.7%.

### 4.3. Temporal Evaluation of the Classification Algorithm

Finally, the 29 daily sequences that were initially chosen were tested with the proposed Smoke Classification with Machine Learning Algorithms, namely using the soft Voting

Algorithm that showed the best performance. Table 4 shows that from the 10,333 total detections, we manually verified only 145 TP from the initial detection and 10,188 FP. It is noted that only the actual detections are considered for the analysis in Table 4, so no values for FN and TN were considered during the first step. All the metrics for each dataset are shown in Table S2. After the second step, 41 of out of the 145 True Positives were rejected (reduction of 28%) by the new algorithm, while it was able to reject 9777 out of the 10,188, which is a reduction of 96.0%. Although the reduction of True Positives is considerable, it is noted that no sequences of fires were lost, so the calculated DTR was still one for each sequence, since at least one image was detected from all of the confirmed occurrences of forest fires.

**Table 4.** Overall metrics of the 29 daily datasets analyzed.

| Total Detections | 10,333 | |
|---|---|---|
| **Total Images** | **174,703** | |
| | **After First Classifier** | **After Second Classifier** |
| TP | 145 | 104 |
| FP | 10,188 | 411 |
| TN | 0 | 9777 |
| FN | 0 | 41 |

Table 5 presents the final results of the detection times, namely with the best object detection algorithm, namely "FRCNN-5000- it.-WUP" with version 3AE, which is defined as the 'First Classifier' [19] and the comparison with the algorithm proposed in this paper, along with the previous results referenced by a previous study [45]. As was expected, the elapsed time for the first detection has increased, because all bounding boxes arise are still created by the 'First Classifier' [19] and then confirmed or rejected by the Machine Learning algorithm.

As can be seen in Table 5, the average of the first detection for all of the previously analysed sequences (1–24) went from 6.3 to 10.9 min, which is still an acceptable value for an early detection. Additionally, 8 new sequences were added to this study from the HPWREN database [47].

When analyzing all the 32 sequences, the elapsed time went from 6.0 to 9.6 min, showing that this newly implemented algorithm is still able to be used in different acquisition conditions and environmental parameters, such as the ones encountered in the diverse outdoor samples of the HPWREN database. It is important to note that these times were manually checked with the first time the smoke column was visible in the picture. There were two fires that had a large increase in the detection time, namely the 20190610_FIRE_bh-w-mobo-c dataset (which went from 5 to 33 min) and the 20200822_BrattonFire_lp-e-mobo-c (from 5 min to 28 min), where the Machine Learning algorithm rejected the first detection and the object detection algorithm did not create any more bounding boxes. It is also important to note that two fires had their initial detections rejected and did not produce any further detection, showing a Not Detected (N.D.) in Table 5.

Additionally, we also calculated the detection time of the previously analyzed 75 images sequences (with the prior and the posterior five frames counted from the occurrence of a fire, as described in the Methodology and Database section). A detection time of 6.5 min was obtained with the newly proposed method (with 64 detections out of 75), which remains an acceptable value considering the value of 5.3 min and 67 detections of the first phase algorithm, taking into account the number of False Positives that this new algorithm manages to remove.

Finally, it is also important to calculate the DTR metric, as defined in the Performance Evaluation subsection. From the 32 analyzed sequences in Table 5, 30 had a missing fire. From the 29 newly acquired dataset, only eight had positive smokes and all eight were still detected, and from the 75 image sequences, 64 were detected, resulting in a DTR of 88.7%

with the proposed algorithm, reduced from 93.0% with just the object detection algorithm. It is noted that this small reduction was always expected, while this new algorithm was shown to eliminate over 95% of the false positives in the 29 daily sequences (reducing from an initial number of 10188 FP to only 411 in over 174,703 analyzed images, which in global terms means that the previously detected False Positive rate of 5.8% for the 29 sequences was reduced to 0.24% in the 29 analyzed images.

**Table 5.** Daytime fire detection time of smoke sequences extracted from HPWREN database. The best results for each case study are marked in bold.

| Video Name | Time Elapsed (min) | | |
| --- | --- | --- | --- |
| | Method [45] | Results with Proposed Classifier | Results with First Classifier [19] |
| Lyons Fire | 8 | **5** | **5** |
| Holy Fire East View | 11 | 3 | **2** |
| Holy Fire South View | 9 | 2 | **1** |
| Palisades Fire | **3** | 7 | 5 |
| Palomar Mountain Fire | 13 | 18 | **10** |
| Highway Fire | **2** | 4 | **2** |
| Tomahawk Fire | 5 | 5 | **3** |
| DeLuz Fire | **11** | 22 | 16 |
| 20190529_94Fire_lp-s-mobo-c | N.A. [1] | **3** | **3** |
| 20190610_FIRE_bh-w-mobo-c | N.A. | 33 | **5** |
| 20190716_FIRE_bl-s-mobo-c | N.A. | **18** | **18** |
| 20190924_FIRE_sm-n-mobo-c | N.A. | 8 | **7** |
| 20200611_skyline_lp-n-mobo-c | N.A. | 6 | **4** |
| 20200806_SpringsFire_lp-w-mobo-c | N.A. | 1 | **1** |
| 20200822_BrattonFire_lp-e-mobo-c | N.A. | 28 | **5** |
| 20200905_ValleyFire_lp-n-mobo-c | N.A. | 14 | **3** |
| 20160722_FIRE_mw-e-mobo-c | N.A. | N.D. [2] | **5** |
| 20170520_FIRE_lp-s-iqeye | N.A. | 10 | **2** |
| 20170625_BBM_bm-n-mobo | N.A. | N.D. | **21** |
| 20170708_Whittier_syp-n-mobo-c | N.A. | 9 | **5** |
| 20170722_FIRE_so-s-mobo-c | N.A. | 16 | **13** |
| 20180504_FIRE_smer-tcs8-mobo-c | N.A. | 16 | **9** |
| 20180504_FIRE_smer-tcs8-mobo-c | N.A. | 3 | **3** |
| 20180809_FIRE_mg-w-mobo-c | N.A. | 8 | **2** |
| 20200822_BrattonFire_lp-s-mobo-c | N.A. | 3 | 3 |
| 20200905_ValleyFire_pi-w-mobo-c | N.A. | 6 | 6 |
| 20200930_BoundaryFire_wc-e-mobo-c | N.A. | 1 | 1 |
| 0200930_inMexico_lp-s-mobo-c | N.A. | 10 | 10 |
| 20200808_OliveFire_wc-e-mobo-c | N.A. | 5 | 5 |
| 0200905_ValleyFire_sm-e-mobo-c | N.A. | 10 | 10 |
| 20200813_Ranch2Fire_wilson-e-mobo-c | N.A. | 4 | 4 |
| 20200930_inMexico_om-e-mobo-c | N.A. | 10 | 3 |
| Mean $\pm$ sd for 1–8 | 7.8 $\pm$ 3.8 | 8.3 $\pm$ 5.5 | 5.5 $\pm$ 3.8 |
| Mean $\pm$ sd for 1–24 | N.A. | 10.9 $\pm$ 6.3 | 6.3 $\pm$ 4.2 |
| Mean $\pm$ sd for 1–32 | N.A. | 9.6 $\pm$ 6.0 | 6.0 $\pm$ 3.8 |

[1] Not available; [2] Not detected.

The above results will impact the number of occurrences that need to be manually confirmed by the Forest Protection Services after the alarm is initially given. It is important to account that future studies about dynamically changing the threshold levels of the classification system, based for example on knowledge about current weather conditions and fire severity classifications, could be done to reduce the number of actual smoke columns that are rejected by the Machine Learning algorithm.

## 5. Conclusions

A deep learning object detection model for the detection of forest fires and their smoke plumes was previously implemented on the Detectron2 platform [19]. That system was recently deployed in two Portuguese locations (Valongo and Proença) for real-time acquisitions, and a daily analysis found an average of around 6% of false alarms, with some days having over 100 warnings in 4000 images. Because of the identification of persistent FP, this study was carried out to upgrade the system with a new algorithm comprised of an object segmentation block of the bounding box and a feature extraction step, a final block with an implemented AI algorithm capable of differentiating the detected features of the segmented objects, reducing the number of FP while maintaining a high accuracy. After the manual segmentation, 29 daily training sessions, with 174703 images, were used to optimize the instance segmentation model, showing an Average Precision metric of the same level, as reported in the literature [46].

Afterwards, several features were extracted from the segmented objects to be trained and tested as inputs of a classification algorithm. A soft voting algorithm was implemented aggregating the three best performing algorithms, namely the MLP classifier (without PCA), the XGBoost classifier (PCA with 12 features), and the Bagging classifier (PCA with nine features). This voting algorithm obtained the best accuracy out of all the parameters and single algorithms (84.60%). This voting model was evaluated by splitting the dataset into 10 consecutive folds, without shuffling, finally obtaining an F1-Score of $91.8 \pm 0.9\%$, which gives high confidence for our classification algorithm.

This proposed model was finally also tested using the HPWREN database [47] and compared with an existing study for the assessment of a smoke detection algorithm for some of the examples present in the database reported in [45]. While their reported time of 7.8 min was superior to the one detected by our previous algorithm (5.5 min), it was noticed that the newly proposed algorithm had a slightly superior time (of 8.3 min), which was mainly due to the DeLuz Fire dataset. It is noted that this increase of time was the expected behavior, and the reduction of the False Positive rate by over 95% clearly outweighs the sacrificed time for the first detection. The HPWREN database has a different geographic setting (Portugal vs. South California) and uses different hardware, which also highlights that this system is able to adapt to novel conditions.

Using 75 sequences, a mean time of 6.5 min was calculated, compared to 5.3 min of the previous algorithm. The increase in the time for the first detection is still considered appropriate for an early detection system, which is capable of warning the fire protection services to rapidly accept or reject the detection and carry out their activities.

It is noted that while the biggest limitation identified during the previous paper [19] was corrected by decreasing the FAR (rejecting over 95% of the false positives), the time for the first detection and the number of smokes detected had a non-negligible decrease. This decrease is in norm with specifications provided by the forest protection services, which appreciate the daily reduction in the number of alarms, as these still need to be manually confirmed, and on a larger scale (e.g., several systems implemented in real time), they could start to be ignored by the human operator.

In the future, it might be interesting to see how different locations (e.g., urban area) can affect this detection, as different types of objects can be detected by the first classifier, and their features could be rejected. A further study could also implement a temporal evolution analysis of the shape of the smoke column, as it might provide additional help in distinguishing it from other objects such as clouds or smog.

**Supplementary Materials:** The following supporting information can be downloaded at: https://www.mdpi.com/article/10.3390/rs14112701/s1, Table S1: Chosen Hyper-parameters for each tested Case, Table S2: Performance Metrics Breakdown for each Dataset.

**Author Contributions:** Conceptualization, all authors; methodology, L.M., F.G.-F. and R.V.d.A.; software, L.M., F.G.-F and R.V.d.A.; validation, L.M. and F.G.-F.; formal analysis, L.M. and F.G.-F.; investigation, L.M. and F.G.-F.; resources, H.G. and P.V.; data curation, L.M. and F.G.-F.; writing—

original draft preparation, L.M. and F.G.-F.; writing—review and editing, all authors; visualization, L.M. and F.G.-F.; supervision, H.G. and P.V.; project administration, H.G. and P.V.; funding acquisition, H.G. and P.V. All authors have read and agreed to the published version of the manuscript.

**Funding:** This research has been supported by project POCI-01-0247-FEDER-038342 from the COM-PETE 2020 program.

**Data Availability Statement:** The data presented in this study are available on request from the corresponding author. The HPWREN data are publicly available from the High-Performance Wireless Research and Education Network website.

**Acknowledgments:** The authors would like to thank Future-Compta S.A for supporting this research work. This project has been supported by project POCI-01-0247-FEDER-038342 from the COMPETE 2020 program.

**Conflicts of Interest:** The authors declare no conflict of interest. The funders had no role in the design of the study; in the collection, analyses, or interpretation of data; in the writing of the manuscript; or in the decision to publish the results.

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
