# Peer review of "Real-Time Integration of Segmentation Techniques for Reduction of False Positive Rates in Fire Plume Detection Systems during Forest Fires"

_remotesensing, doi:10.3390/rs14112701_

Round 1

Reviewer 2 Report

The paper is OK.

Reviewer 3 Report

As stated by the authors, this paper is a direct continuation of a previous work. It presents new developments to a previously implemented system for the classification of smoke columns with object detection and deep learning based approach. The aim of this work is to lessen these drawbacks by developing and validating a solution that reduces false positives, thus proposing an hybrid model that combines several strategies, such as deep learning algorithms with instance segmentation, feature extraction and traditional machine learning classification techniques. The subject of this paper is important for the fire detection for the governmental offices due to the social, environmental and economic consequences of uncontrolled fires. The main impact of the results obtained by this proposed method is that when applied to real world systems, every alarm still need to be confirmed by a manual operator from the forest protection services, and persistent false positives could lead to a human tendency to start ignoring some of those alarms. The results seem to be reasonable and are supported by the data and the method used.

Some specific comments

L.42- ranging from from  ---   ranging from

L.226-with the the  ---   with the

L.248- backbone combinedwith the FPN. and initialzedweights  ----  backbone combined with the FPN. and initialized weights

L.268- in images acquired by. ?

L.322- time elapsed to to detect  ---   time elapsed to detect

L.325-showing the the smokes  ---   showing the smokes

Figure 5- (C) was was later  ---   (C) was later

L.495- After the the best  ---  After the best

L.500- This features were  ---  These features were

Conclusions Section is too long. It should be more synthetic highlighting the main results. Most of the information on this section could be moved to Results and Discussion Sections.

Reviewer 4 Report

This manuscript is a continuation of Guede-Fernandez et al., 2021 published work. The authors improved the methodology by decreasing the false positive rate of fire and smoke detection. A comprehensive analysis of 6000 images daily was performed to show the robustness of the algorithm. I recommend the manuscript for publication after a round of minor revision.

Please see some minor comments below:

  1. There are five occasions where an extra "the" article is inserted. Check the lines 27, 226, 325, 495, 501.
  2. Line 12" check ", This model"
  3. Line 66 change to "the largest use is the application"
  4. Line 76: consider adding FireScout AlcheraX for the fuller picture
  5. Line 169 consider removing "aslo"
  6. Line 179 check "added Due to this"
  7. Line 148 check "combined with the FPN. and initialized"
  8. Line 462 change "an" to "a"
  9. Line 479. Please elaborate on "because of this situation". 

Author Response

Please see the attached response to the reviewer in PDF.

Round 2

Reviewer 1 Report

The author's response letter has better solved my concerns, but I still think the introduction section should be more short and concise.

Author Response

We would like to thank the reviewer for his thorough review of our manuscript, improving significantly the quality of our Manuscript. We have made additional revisions to the Introduction Section. Some parts of the Introduction were removed (see some examples):

  • the "Lightning discharges are accountable for only 10% of the wild fire accidents while 22 human undertakings account for the rest of the events [3]." phrase was removed
  • the paragraph that starts with "Forest wildfires are normally characterized" was joined with another paragragh and was shorttened.
  • the paragrapgh that starts with "Recent reviews have shown the long term importance of investing" was extensively reviewed to be more concise

All the changes to the Introduction Section are highlighted by the new submitted PDF.

We believe that with this revisions acknowledge the improvements suggested by the reviewer. 

Reviewer 3 Report

As stated by the authors, this paper is a direct continuation of a previous work. It presents new developments to a previously implemented system for the classification of smoke columns with object detection and deep learning based approach. The aim of this work is to lessen these drawbacks by developing and validating a solution that reduces false positives, thus proposing an hybrid model that combines several strategies, such as deep learning algorithms with instance segmentation, feature extraction and traditional machine learning classification techniques. The subject of this paper is important for the fire detection for the governmental offices due to the social, environmental and economic consequences of uncontrolled fires. The main impact of the results obtained by this proposed method is that when applied to real world systems, every alarm still need to be confirmed by a manual operator from the forest protection services, and persistent false positives could lead to a human tendency to start ignoring some of those alarms. The results seem to be reasonable and are supported by the data and the method used. The manuscript has been improved by the authors following the reviewers comments and suggestions.

Author Response

We would like to thank the reviewer for his revision of the paper and for his comments. We made some final changes in the Introduction section according to the first reviewer in order to make it more concise as highlighted by the submitted PDF: